# A Collaborative Approach to Support Participation in Physical Recreation for Preschool Students with Multiple Disabilities: A Case Series

Heather L. Brossman [1,*], Lisa A. Chiarello [1], Robert J. Palisano [1] and Kimberly D. Wynarczuk [2]

1 Department of Physical Therapy and Rehabilitation Sciences, Drexel University, Philadelphia, PA 19104, USA; lisa.chiarello@drexel.edu (L.A.C.); robert.j.palisano@drexel.edu (R.J.P.)
2 School of Rehabilitation Sciences, Moravian University, Bethlehem, PA 18018, USA; wynarczukk@moravian.edu
* Correspondence: heather.brossman@drexel.edu

**Abstract:** *Aims*: To evaluate a collaborative participation-based therapy approach for two preschool students with multiple disabilities from the experience of Individualized Education Program (IEP) teams, highlighting the perspective of the physical therapist. *Methods*: The phases of collaborative participation-based therapy were implemented: (a) collaborative relationships were developed and supported within the IEP teams, (b) collaborative meaningful physical recreation goals were developed for participation at school, (c) strengths and needs assessments using the "*Collaborative Process for Action Plans to Achieve Participation Goals*" were conducted with IEP teams, (d) participation-based interventions were provided, and (e) goal achievement and processes were evaluated. The use of technology for collaboration was encouraged. The physical therapist kept intervention logs and wrote reflective journal entries. Interventions adhered to COVID-19 regulations. Goal achievement was measured using Goal Attainment Scaling. IEP team members completed questionnaires on their experiences. *Results*: One student met their goal expectation, and one student exceeded their goal expectation. The students were engaged, and IEP team members' experiences were positive. Conclusions: The use of participation-based therapy is meaningful, feasible, and acceptable to IEP teams. Team collaboration and flexibility were instrumental to successful implementation. Strategies to promote effective communication and the use of technology would support a participation-based therapy approach.

**Keywords:** preschool-based physical therapy; multiple disabilities; physical therapy; related services; participation-based approach; collaboration; physical recreation

## 1. Introduction

The inclusion of children with disabilities in preschool is mandated by the Individuals with Disabilities Education Act [1]. Inclusion is not just placing a student in the least restrictive educational environment but enabling the student to establish a sense of belonging, learn with peers, and develop social relationships [2]. The complex combination of intellectual, behavioral, communication, and motor difficulties for preschool students with multiple disabilities may restrict participation in school activities [3].

Young children in preschool learn through play by communicating, making choices, exploring the environment, and participating with peers [4]. Play and physical activities are a vital part of preschool education, promoting health and wellness, teaching social skills, and advancing motor abilities [4]. Supporting young children with multiple disabilities to participate in movement activities may advance their motor skills, promote friendships, expand communication, and enable them to discover what they enjoy doing [3]. The discovery of what they enjoy doing at a young age may contribute to the development of self-determination, which is important for more autonomous living and

is a focus of special education [5]. For young children with disabilities, preschool personnel and family members work collaboratively to develop the student's Individualized Education Program (IEP). The IEP directs student learning and participation in preschool activities. The successful inclusion of preschool students with multiple disabilities in school activities requires strong communication and collaboration among the members of the IEP team [6,7]. Collaboration utilizes the collective knowledge, unique skills, and expertise of each team member with families as equal partners to develop child-focused goals that are transdisciplinary and meaningful. Publications in the literature on team collaboration for preschool physical recreation are limited to brief notations on collaborative practices to support students with severe disabilities [6,7]. Although these studies included physical recreation activities to illustrate teaming, they did not describe the interventions provided to support physical recreation.

Parents of preschool-aged children with mobility limitations have reported that their children participate less frequently in family and recreation activities than parents of children without these conditions [8]. Parents of young children with disabilities have reported that the preschool/daycare environment was less supportive for their children and that their children participated less frequently and were less involved in group learning, socializing with friends, and field trips/events compared to reports by parents of children without disabilities [9]. Di Marino et al. [10] identified functional difficulties, such as limitations in gross motor abilities and communication and difficulty in regulating behavior, as barriers to the participation of young children with disabilities in the preschool/daycare environment. Parents expressed a desire to improve participation for their young children with disabilities in preschool/daycare and reported using strategies such as task and environmental modifications to improve children's behavior and facilitate social interaction [9].

Educational approaches that embed instructions in the activities of the school day, such as activity-based instruction, are aligned with current pediatric rehabilitation practices, such as goal-directed and participation-based interventions, including participation in physical activity [11–17]. Evidence exists to support participation-based therapy approaches; however, these studies primarily include children aged 5–18 years and focus on participation in community recreation [13,15–17].

The studies on community recreation in the literature do not address the outcomes of interventions to promote young children under the age of five years old with multiple disabilities and significant limitations in movement partaking in physical activity. Additionally, the studies on preschool service delivery do not report the roles and experiences of an IEP team inclusive of families, educators, and all related service providers. The purpose of this prospective case series is to describe a participation-based therapy approach to support participation in preschool physical recreation activities for two students with multiple disabilities. The report showcases collaborative goal setting, service delivery processes, student outcomes, and the IEP team's experiences with and perspectives on this innovative approach. The support and services provided by the physical therapist (PT) for goal achievement and service adaptation made during the COVID-19 pandemic are highlighted.

## 2. Materials and Methods

### 2.1. Overview of the Adapted Participation-Based Approach for the Preschool Context

2.1.1. Measures and Documents

Goal Attainment Scaling (GAS). GAS, an individualized criterion-referenced measure of goal achievement, was the primary student outcome. GAS quantifies a goal over five levels of outcomes, from −2 (current level of function) to +2 (a level that exceeds expectations), with 0 being the expected level of achievement. The levels of goal attainment should have clinically meaningful, equal intervals between them. GAS has good interrater reliability, is sensitive to change over time, and is responsive to minimally clinically relevant changes over a wide variety of ages and diagnoses [18,19]. GAS has been used to document educational improvements for students with multiple disabilities [20].

The Collaborative Process for Action Plans to Achieve Participation Goals. The *Collaborative Process for Action Plans* to Achieve Participation Goals (referred to as *Collaborative Process for Action Plans* for brevity) for home- or community-based participation goals [21] was adapted for educational teams serving young children (3–6 years of age) with multiple disabilities in a preschool setting (Supplement S1). This process documents a strengths-based assessment of attributes related to the child, family, teacher, related service providers (RSPs), and environment for the development of an action plan as it relates to a preschool-based physical recreation participation goal. The *Collaborative Process for Action Plans* was evaluated by pediatric rehabilitation therapists [21]. Overall, 93% and 89% of the attributes met the criterion for consensus agreement during two evaluations, supporting content and construct validity. Pilot testing indicated the measure promoted engagement and captured multiple viewpoints in the development and commitment to an action plan, providing evidence for the measure's usefulness [21].

Intervention Log. This researcher-developed documentation form (Supplement S2) details components of participation-based services and supports provided directly to the student and on behalf of the student, including communication and collaboration with IEP team members. This log was informed by the literature related to participation-based interventions [11,13], school-based practice competencies [22], and the experiences of the PT and research team.

Student Engagement. This researcher-developed questionnaire (Supplement S3) has two Likert-scaled questions (five-point response options ranging from "not at all" to "really a lot") and four open-ended questions for the teacher to share their perceptions of the child's engagement in and enjoyment of physical recreation activities [13].

Experience Questionnaires. Experience questionnaires (Supplement S4) were adapted from Chiarello and Palisano [23] to obtain the family's, teacher's, and RSP's perspectives regarding participation-based therapy. The questionnaires have closed-ended questions related to the fidelity and usefulness of the components of participation-based therapy and open-ended questions to solicit one's perspectives on what they like and dislike about the approach, as well as the benefits, barriers, and facilitators of the approach.

### 2.1.2. Procedures

The first author, the PT on the IEP teams, served as the lead for the processes involved in this study. This study followed the five phases of participation-based therapy, as described by Palisano et al. [13]. The objective of phase one is to develop collaborative relationships. The school-based teams in this study had previously established collaborative relationships, which were supported through the process. In phase two, a goal was collaboratively identified for the student during a phone call with the family, teacher, and PT. The goals considered the physical recreation interests of the child and family, the educational curriculum, and the materials available during the activities of the school day. GAS levels were developed, reviewed, and finalized with the second author to ensure they met GAS criterion and subsequently shared with the IEP teams [24]. The GAS levels were posted on a wall in the classroom. During phase three, a virtual collaborative meeting was held with the IEP team to explore the attributes of the child, family, teacher, RSPs, and environment related to the physical recreation goal from a strengths-based approach using the *Collaborative Process for Action Plans*. The PT facilitated the discussion, and the team developed an action plan which identified outcomes to achieve the goal and the responsible team members. In phase four, participation-based interventions were provided. Intervention strategies emphasized communication and collaboration among the team, the identification and application of resources, and instruction and guided practice with the children in the activity during the routines of the school day. Families, teachers, and RSPs were encouraged to utilize technology to provide their perspectives and share videos and photos of the student practicing the activity. The PT documented services and supports on the intervention log and kept notations in a journal. Phase five involved the evaluation of processes and outcomes, during which the families, teachers, and RSPs completed experi-

ence questionnaires. The teacher completed the engagement questionnaire, and the teacher and PT collaboratively rated goal achievement.

### 2.1.3. Data Synthesis

The first two authors documented their assumptions regarding preschool service delivery before participants were enrolled in the study. Recordings of the *Collaborative Process for Action Plans* meetings were viewed and transcribed. A detailed narrative description of the cases, including student engagement and GAS outcome scores, and a tabular presentation of attributes and action plans were synthesized. Researchers calculated the proportion of weeks during which the intervention types were provided. Intervention data and quantitative participants' experience responses were visually presented in tables to provide comparisons across cases and participants, respectively. Narrative responses from items on the questionnaires were jointly summarized by the first two authors in a consensus-agreement process using the participants' own words to ensure trustworthiness. All positive and constructive feedback was described. The synthesis of the cases considered the reflections noted in the PT's journal and contextual attributes of the team and preschool environments.

### 2.2. Case Descriptions

### 2.2.1. Participants and Setting

This study was approved by the Drexel University's Institutional Review Board. The participants were two preschool students with multiple disabilities and their IEP teams. For this case series, the students were recruited as a sample of convenience from the caseload of the PT and given pseudonyms. Inclusion criteria were a preschool student with multiple disabilities who had an IEP; no scheduled surgery, breaks, or plans to move during the study period; and an interest in physical recreation participation (as expressed by the IEP team). The mothers, as the family-chosen participant in the study, provided consent. Once the mothers consented to the study, the lead teacher and RSPs on the team were recruited, and they consented to participating in the study. The teacher and mothers had access to smartphones to ensure that they could send and receive text messages and videos.

The students were in the same self-contained special education classroom with a special education teacher, instructional assistant, classroom nurse, and assigned RSPs who had worked as a team for the previous five school years. The classroom teacher, PT, occupational therapist (OT), and speech language pathologist (SLP) were the same for both students. One of the students had a vision teacher. The demographic data of the study participants can be found in Table 1. The members of the school-based team were all white women between 42 and 58 years of age and educated at the master's to doctorate level with 15–30 years of professional experience. The adapted resources available to students in the classroom included communication devices, standers, and gait trainers.

**Table 1.** Participant demographic information.

| Student Characteristics | Jake | Luke |
|---|---|---|
| Gender | Male | Male |
| Age (years) | 3.5 | 3.8 |
| Race | Asian Indian | Caucasian |
| Diagnosis | Gene Mutation DD * | Cerebral Palsy |

**Table 1.** *Cont.*

| Student Characteristics | Jake | Luke | |
|---|---|---|---|
| Classification systems | NA | GMFCS * | level V |
| | | MACS * | level V |
| | | CFCS * | level V |
| School-based therapies | PT *, OT *, SLP *, Vision * | PT, OT, SLP | |
| Type of school environment | Specialized classroom | Specialized classroom | |
| Outpatient therapies | PT, OT | PT, OT, SLP | |

| Maternal Characteristics | Jake | Luke |
|---|---|---|
| Gender | Female | Female |
| Age (years) | 35 | 35 |
| Race | Asian Indian | Hispanic/Latino |
| Marital status | Married | Married |
| Employment | Not employed | Not employed |
| Education | Master's degree | High school graduate |

| School Personnel Characteristics | Teacher * | PT | OT | SLP | Vision |
|---|---|---|---|---|---|
| Gender | Female | Female | Female | Female | Female |
| Age (years) | 58 | 50 | 55 | 42 | 50 |
| Race | Caucasian | Caucasian | Caucasian | Caucasian | Caucasian |
| Education | Master * | Doctorate | Master | Master | Master |
| Years in practice | 15 | 23 | 30 | 18 | 20 |
| Years in preschool setting | 8 | 6 | 10 | 8 | 20 |

* Note: Master = master's degree, DD = developmental delay, Teacher = classroom teacher, PT = physical therapist, OT = occupational therapist, SLP = speech language pathologist, Vision = vision teacher, GMFCS = Gross Motor Function Classification System, MACS = Manual Ability Classification System, CFCS = Communication Function Classification System.

Jake. At the start of the study, Jake was a 3.5-year-old boy. He has been diagnosed with a rare genetic mutation, cortical visual impairment, seizure disorder, and developmental delay. Jake has a happy, calm disposition. He uses nonverbal behaviors to express his desires by looking at and reaching for materials. He is not consistent with yes or no responses but can demonstrate refusals by vocalizing or turning away. Jake's main mode of transportation is being pushed in a wheelchair. He can walk in a gait trainer with bilateral articulating modified foot orthoses (MAFOs) for up to 150 feet with maximal assistance. Jake did not demonstrate a hand preference. He needs support and guidance to bring two hands together and to cross midline.

Luke. At the start of the study, Luke was a 3.8-year-old boy. He has been diagnosed with spastic quadriparetic cerebral palsy (CP). Luke is engaging but often cries and is hard to be console. He is visually observant, uses nonverbal communication methods that are difficult to understand, and is not able to communicate yes or no. Luke's main mode of transportation is being pushed in a wheelchair. He can stand in a gait trainer with bilateral MAFOs and head support and requires maximal physical assistance to progress his lower extremities to take steps. Luke has strong extensor tone. He moves his arm from the shoulder to hit a switch, has limited hand function, and wears bilateral hand splints. Luke is classified as level V on the Gross Motor Function, Manual Abilities, and Communication Function Classification Systems [25–27].

### 2.2.2. Setting Goals

Each student's physical recreation goal for the study was aligned with their IEP goals (Supplement S5). The IEP goals focused on communication, motor function, and social skills. The students' GAS levels for their physical recreation goal are listed in Table 2.

**Table 2.** Goal Attainment Scaling (GAS) levels for Jake and Luke's recreational goals.

| Scale | Jake Goal Description: Basketball | Luke Goal Description: Bowling |
|---|---|---|
| | During classroom gross motor time with a peer, Jake stands in a gait trainer (lateral supports, no sling) to observe a colorful ball (placed on a tray with a black background without complexity). | During classroom gross motor time with a peer, Luke stands in a gait trainer with head support to utilize a ramp to push the ball down the ramp: ** |
| −2 = Current Level (Baseline) | Jake touches the ball with **one hand to swipe the ball** off the tray (**with physical prompting**) and into a bucket held by an adult, awaits the completion of their peer's turn, **utilizes a switch to request "my turn" (with physical prompting)**, and swipes the ball off the tray with one hand and into the bucket held by an adult (with physical prompting) for a second time. * | With **full physical prompting** (Luke is contributing physical effort; **adult is providing support for nearly 100% of process**), Luke **utilizes a switch (with full physical prompting)** to request "my turn" and repeatedly pushes the ball down the ramp with full physical prompting. * |
| −1 = Less than Expected | Jake touches the ball **with both hands to roll the ball** off the tray and into a bucket held by an adult (**with physical prompting**), awaits the completion of their peer's turn, **utilizes a switch to request "my turn"** (**with physical prompting**), and rolls the ball off the tray with both hands and into the bucket held by an adult (with physical prompting) for a second time. * | With **partial physical prompting** (Luke is contributing physical effort; **adult is supporting 50% of process**), Luke **utilizes a switch (partial physical prompting 50%)** to request "my turn" and repeatedly pushes the ball down the ramp with partial physical prompting (50%). * |
| 0 = expected | Jake touches the ball **with both hands (with verbal prompting)** to roll the ball off the tray and into a bucket held by an adult, awaits the completion of their peer's turn, **utilizes a switch (with verbal prompting)** to request "my turn", and rolls the ball off the tray with both hands and into the bucket held by an adult (with verbal prompting) a for a second time. * | With **partial physical prompting** (Luke is contributing physical effort; **adult is providing support for 25% of process**), Luke utilizes a switch (**partial physical prompting 25%**) to request "my turn" and repeatedly pushes the ball down the ramp with partial physical prompting (25%). * |
| +1 = more than expected | Jake touches the **ball with both hands (with a model)** to roll the ball off the tray and into a bucket held by an adult, awaits the completion of their peer's turn, **utilizes a switch (with verbal prompting)** to request "my turn", and rolls the ball off the tray with both hands (with a model) and into the bucket held by an adult for a second time. * | With **verbal prompting and an appropriate wait time of up to 90 s**, Luke utilizes a switch (**verbal prompting and an appropriate wait time up to 90 s**) to request "my turn" and repeatedly pushes the ball down the ramp with verbal prompting. * |
| +2 = much more than expected | **Jake picks up the ball with both hands, places the ball (with a model)** into a bucket held by an adult, awaits the complete of their peer's turn, **utilizes a switch independently (with an appropriate wait time of up to 90 s)** to request "my turn", picks up the ball with both hands, and places the ball (with a model) into the bucket held by an adult for a second time. * | **Independently** and with an **appropriate wait time of up to 90 s**, Luke utilizes a switch (**independently**) to request "my turn" and repeatedly pushes the ball down the ramp independently. * |
| Final GAS score | 0 | +1 |

Note: * Three out of four opportunities during weekly data collection and reported by the physical therapist; ** position for Luke modified from sitting to supported standing based on his mother's input during the *Collaborative Process for Action Plans to Achieve Participation Goals* meeting. The bold typeface is for clarity in differentiating levels of the GAS.

Jake's Goal. During the goal-setting meeting, Jake's mother was focused on him improving his standing time, walking endurance, and two-handed abilities. She also wanted to identify a game Jake could play with his brother. The team agreed that Jake loved ball play and that an adapted basketball game would be engaging. The game consisted of standing in a gait trainer, moving a ball into a bucket, and using an adapted switch to request another turn. Participation in this game included components of his IEP goals, such as the use of a switch, bimanual abilities, standing endurance, turn-taking, and preacademic knowledge of directionality (e.g., in/out).

Luke's Goal. During the goal-setting meeting, Luke's mother shared that he was aware of what was going on around him and was trying to express himself. The team selected an adapted bowling game as the family lives near a bowling lane, and the teacher identified many preschool concepts that could be taught through the game. The game consisted of pushing a ball down a ramp while seated in his wheelchair and using an adapted switch to request another turn. Participation in this bowling activity included components of his IEP goals, such as reaching, the use of a switch for communication, and preacademic knowledge of counting.

### 2.2.3. Discussing Strengths and Needs to Establish an Action Plan

The *Collaborative Process for Action Plans* analysis of attributes for both students can be found in Supplements S6 and S7. The environmental strengths that were highlighted by the teams were the classroom resources and incorporation of a movement-oriented curriculum with supportive adults. For the action plans, the teams emphasized choice making, communication, self-initiated movement, and playfulness. The PT was responsible for equipment, positioning, adaptations to the games, and collaborating with other team members to enable practice. The SLP collaborated with the classroom teacher and PT for the picture exchange and voice output switch to state "my turn". At the end of the meeting, Luke's mother stated, "I am very happy to be honest with you about this time that we had together, because I can see that everybody knows him. As his mom, I had a lot of concern about that, but now I can see you all have a good picture of him, and you know him, and I am really happy with that".

Jake's Attributes and Action Plan. The key attributes for Jake in support of his goal included his interest in ball play and his family's desire for him to play with others. The areas of needs included seizure precautions, the safety of his G-Tube, standing endurance, and enhancing visual attention. The action plan to support Jake's basketball goal (Table 3) also included collaboration with his vision teacher and homecare nurse.

**Table 3.** Jake's action plan.

| Outcomes to Achieve the Goal (Child, Family, Teacher, Related Service Providers and Environment Attributes) | Person Responsible: Actions, Strategies, and Procedures |
|---|---|
| Optimize participation in basketball game with peers through team adoption of adaptations and environmental support | **Physical therapist:** <br> • The PT will attend gross motor time and leisure time in class; <br> • The PT will provide access to black buckets and a ball that is saturated in its color to optimize attention; <br> • The PT will increase the amount of time that the gait trainer is used to improve cardiovascular endurance and standing tolerance; <br> • The PT will record a video of Jake playing the adapted game of basketball and share the video with all team members; |

**Table 3.** *Cont.*

| Outcomes to Achieve the Goal (Child, Family, Teacher, Related Service Providers and Environment Attributes) | Person Responsible: Actions, Strategies, and Procedures |
|---|---|
| Optimize participation in basketball game with peers through team adoption of adaptations and environmental support | **Physical therapist:**<br>• The PT will teach classroom staff (teacher, instructional assistant, related service providers, classroom nurse) to set up and play the adapted basketball game, including the correct positioning of Jake in the gait trainer;<br>• The PT will teach the team to place materials on the tray to ensure they are closer than 4 feet in upper fields and move the ball to gain Jake's visual attention before the game starts;<br>• The PT will ensure that peers are up in a standing position for the teacher, nurse, and instructional assistant to support Jake to play with peers (while adhering to COVID-19 standards);<br>• The PT will develop photo documentation to show proper modified ankle foot orthosis use.<br>**Classroom team members** (teacher, classroom nurse, instructional assistant, related service providers):<br>• The classroom team members will check Jake's G-Tube before, during, after play.<br>**Home nurse:**<br>• The home nurse will add the adapted game to Jake's daily evening schedule and ensure that the materials needed for the game are at home. |
| Play with peers and make the game playful | **Physical therapist:**<br>• The PT will schedule sessions to provide services during gross motor and leisure time in the classroom.<br>**Classroom team members** (teacher, instructional assistant, related service provid. ers, classroom nurse):<br>• The classroom team members will utilize the gait trainer in classroom during gross motor time with the student and a peer;<br>• They will act excited when the ball drops into the bucket by engaging in playful behavior, such as joyfully saying "two points" (as in basketball points) when the ball goes into bucket;<br>• They will support turn-taking by exchanging pictures or using a switch/sign language (multi-modal) to indicate "my turn" or "your turn" and include peers during the game. |
| Communication utilized to increase engagement and participation | **Speech language pathologist:**<br>• The SLP will provide pictures to signal "my turn", "your turn", "wait", and "ball";<br>• The SLP will share a picture of the actual balls used with the team.<br>**Classroom team** (teacher, nursing, instructional assistant, related service providers):<br>• The classroom team will encourage choices using picture exchange or multi-modal communication before, during, and within the game both in school with adults and peers and at home with family (brother/nurse/mom);<br>• They will also teach sign language with hand-over-hand modeling.<br>**Physical therapist:**<br>• The PT will offer a choice between balls;<br>• The PT will utilize switches during gross motor activities. |

**Table 3.** *Cont.*

| Outcomes to Achieve the Goal (Child, Family, Teacher, Related Service Providers and Environment Attributes) | Person Responsible: Actions, Strategies, and Procedures |
| --- | --- |
| Encourage the use of both hands and crossing midline while participating | **Team:**<br>• The team will offer hand-over-hand assistance for the participant to ensure they reach the ball and place it into the bucket or roll it into the bucket, ensuring movement across the midline. |

Luke's Attributes and Action Plan. At the onset of the meeting, Luke's mother shared how much he disliked sitting in his chair. After a team discussion, the goal was modified to standing in a gait trainer for the game, which pleased his mother. The key attributes supporting Luke's goal included his visual attention to the environment and family interest in bowling. The primary area of need was enabling a comfortable standing position to ensure Luke was calm and engaged during the activity. Luke's action plan (Table 4) highlighted essential preparations and supports for the team to enable Luke to participate in this classroom activity. Bowling pins and a ball were provided for practice at home.

**Table 4.** Luke's action plan.

| Outcomes to Achieve the Goal (Child, Family, Teacher, Related Service Providers and Environment Attributes) | Person Responsible: Actions, Strategies, and Procedures |
| --- | --- |
| Optimize participation in bowling game with peers through team adoption of adaptations and environmental supports | **Physical therapist:**<br>• The PT will schedule sessions to provide services during gross motor and leisure time in the classroom;<br>• The PT will help to facilitate practicing the use of a textured ball without hand splints;<br>• The PT will adapt the game to ensure that Luke's whole arm is being used to push the ball;<br>• The PT will ensure that the pins fall on a mat (not tiled floor) to reduce startle reflex;<br>• The PT will send a set of bowling pins and a ball to the family home;<br>• The PT will set up an area for bowling with an adapted ramp (while adhering to COVID-19 standards) for ease of classroom use;<br>• The PT will find positions of comfort and stability to enable the use of the communication adaptations.<br><br>**Classroom team members** (teacher, classroom nurse, instructional assistant, related service providers):<br>• The classroom team members will implement the use of the gait trainer, stander, and adapted bowling ramp;<br>• They will support peers to bowl together (while adhering to COVID-19 standards);<br>• They will establish taking two turns, like in "real bowling";<br>• They will create a fun way to keep score and introduce mathematical concepts;<br>• They will address barriers to increase the level of attention being paid to the task;<br>• They will exhibit a playful exuberance to increase playfulness, engagement, and decrease crying;<br>• They will check Luke's G-Tube before, during, and after play.<br><br>**Home Nurse:**<br>• The home nurse will add the adapted game to Luke's daily evening schedule and ensure that the materials needed for the game are at home. |

**Table 4.** *Cont.*

| Outcomes to Achieve the Goal (Child, Family, Teacher, Related Service Providers and Environment Attributes) | Person Responsible: Actions, Strategies, and Procedures |
|---|---|
| Support in standing | **Physical therapist:**<br>• The PT will teach the team a range of motion for extremities and trunk before standing;<br>• The PT will ensure the utilization of the gait trainer in the classroom during gross motor time with Luke and a peer;<br>• The PT will support an increase in Luke's standing time while in a calm, alert, and awake state;<br>• The PT will develop photo documentation for proper modified ankle foot orthosis use;<br>• The PT will post Luke's standing program in the classroom. |
| Communication utilized to increase engagement and participation | **Speech language pathologist:**<br>• The SLP will provide pictures to signal for "my turn", "your turn", "wait", and "ball";<br>• The SLP will share a picture of the actual balls used with the team.<br>**Classroom team** (teacher, nursing, instructional assistant, related service providers):<br>• The classroom team will utilize switches during gross motor activities;<br>• They will encourage choices using multi-modal communication before, during, and within the game with a peer;<br>• They will support turn taking via the use of switches, picture exchange, and/or eye gaze for "my turn" or "your turn";<br>• They will demonstrate comfort talking to Luke, as he understands what is being said to him;<br>• They will give Luke an appropriate amount of time to provide a response (up to 90 s). |

2.2.4. Physical Therapy Sessions and Interventions

Jake's Physical Therapy Sessions. Jake participated in 16 physical therapy sessions during 9 weeks of service across a period of 14 weeks. He received 660 min of total direct service time in his classroom; 56% of this time consisted of participation-based interventions related to his physical recreation goal. Due to COVID-19 quarantine requirements, during three sessions, the PT could only be present in the classroom via virtual means. One session was an in-person home-based session.

Luke's Physical Therapy Sessions. Luke participated in 12 physical therapy sessions during 7 weeks of service across a period of 12 weeks. He received 360 min of total direct service time in his classroom; 78% of this time consisted of participation-based interventions related to his physical recreation goal. Due to COVID-19 quarantine requirements, in one session, the PT could only be present in the classroom via virtual means.

Interventions. The most frequent participation-based interventions provided directly to the students by the PT included the provision of instructions, modeling, and guidance to improve the students' physical, behavioral, social, language, and cognitive abilities; supporting child's active engagement and self-initiated movement; including peers; and addressing body structure and function within the context of the goal activity. During the sessions, the PT also frequently provided instructions and encouragement to school team members participating and supporting the child in the activity. Frequent services on behalf of the students to support the goal included collaborating with the IEP team and accessing materials. Frequent methods of collaborating included communicating through technology and sharing a video. A detailed summary of the interventions is provided in Table 5.

Table 5. Summary of percentage of therapy sessions that each component of intervention occurred.

| Description | Jake | Luke |
|---|---|---|
| Number of sessions | 16 | 12 |
| Total service time | 660 min | 360 min |
| Proportion of time provided participation-based interventions | 56% | 78% |
| Proportion of sessions provided virtually | 25% | 8% |
| Direct services | | |
|     Observed child participating/practicing goal with peers | 44% | 42% |
|     Supported active child engagement | 75% | 100% |
|     Supported child-initiated movement | 81% | 100% |
|     Included peers with activity | 56% | 50% |
|     Established activity adaptations and accommodations | 38% | 33% |
|     Established environmental adaptations and modifications | 25% | 17% |
|     Provided instruction, modeling, and practice of goal | | |
|         Physical, behavioral, social, language, cognitive abilities | 69% | 100% |
|         Physical abilities | 19% | 0% |
|         Behavioral and social abilities | 0% | 0% |
|         Language and cognitive abilities | 6% | 0% |
|     Provided instruction to individuals participating and supporting the child in the activity | 50% | 58% |
|     Supported health promotion and injury prevention in context of activity | 13% | 25% |
|     Addressed body structure/function | | |
|         within context of activity | 63% | 92% |
|         not in context of activity | 13% | 50% |
| Student supports and services: On behalf of student | | |
|     Planned activities, adaptations, environmental modifications | 19% | 42% |
|     Accessed materials and resources | 31% | 67% |
|     Shared and discussed resources | 31% | 25% |
|     Coordinated opportunities: scheduling, assistance | 38% | 25% |
|     Provided encouragement for classroom staff in support of the child during the physical recreation goal | 31% | 50% |

## 3. Outcomes

### 3.1. Goal Attainment

Jake achieved the level of expectation for goal achievement, with a GAS score of 0. He stood in a gait trainer to play the game with a peer; with verbal prompting, he used both hands to roll a ball off a tray and into a bucket after utilizing a switch to request "my turn" and then repeated the sequence.

Luke exceeded the level of expected goal achievement, with a GAS score of +1. Luke stood in a gait trainer with head support to play the game with a peer; with verbal prompting, he pushed a ball down a ramp after utilizing a switch to request "my turn" and then repeated the sequence.

### 3.2. Participant Experiences

#### 3.2.1. Student Engagement

For the Likert-scaled items, the teacher reported that both students had "really a lot" of fun and learned "really a lot" from participating in their adapted games. In the open-ended questions, the teacher commented that Jake liked being out of his adapted chair to play with a ball and enjoyed the positive reactions of the staff. The teacher reported that he learned to visually attend, participated actively, and enjoyed hitting the switch to communicate. For Jake, the teacher did not identify any dislikes related to participating in the activity but noted that a barrier to participation was his medical status related to seizures. For Luke, the teacher noted that he would laugh out loud when his ball knocked over the bowling pins and he loved all the attention and praise he received from the school-based team for hitting

the "my turn" switch. The teacher reported that Luke did not always like standing or that he had to do the activity independently. Luke learned that he received attention when he participated. He also learned how to reach his arms, open his hands, and activate the switch to communicate and play the game. The teacher's comments on barriers to Lukes' enjoyment of the activity related to reduced opportunities because of part-time student status, COVID-19 adaptations, and his school schedule.

### 3.2.2. Perceptions of Mothers, Teachers, and Related Service Providers

The mothers, teachers, and RSPs reported that they were moderately to completely involved and very to completely satisfied with the participation-based therapy. They indicated that the team collaborated to meet the students' goals and that the therapy approach enhanced collaboration by a great to very great extent. The team members rated that the students enjoyed participation a great to very great extent, and the components of participation-based therapy were found to be somewhat to extremely helpful. The school-based team members shared positive aspects about the collaborative process and its outcomes, identifying benefits for the student and themselves. The classroom teacher noted that the structured meeting was "excellent as we had dedicated time for a discussion of the child specific to his goal with the family collaborating". The OT stated that it "bonded the IEP team members, especially the parents". The classroom teacher emphasized that sharing the videos of game play with the families showcased the students "excitement and independence of the child outside the home and parents were reassured that their children were safe and enjoying school while developing skills". The mothers had varied experiences. Jake's mother provided less information, noting that the process was carried out in school, but she appreciated her son's engagement in the therapy approach and his accomplishment of learning to play basketball. Luke's mother liked the focus on playing independently and highlighted that her son gained motivation and the benefit of being able to identify preferences. The participants stated that the team members were the biggest facilitator in helping the students achieve their goals. The teacher noted her belief that movement is vital for all students: "they need to move and we need to make it fun and functional". Participants recognized challenging aspects and barriers related to the time taken to discuss the attributes and develop the action plan, the short-time frame for implementation, students' part-time preschool schedule, and the complexity of the students' learning needs and daily routines. Participant Likert-scaled responses regarding their experiences with the participation-based therapy approach are shown in Table 6, and a summary of the responses to the open-ended items are detailed in Supplement S8.

**Table 6.** Summary of the responses to the participant experience questionnaires.

| Question | Jake's Mother | Luke's Mother | Teacher | PT | OT | SLP | Vision |
|---|---|---|---|---|---|---|---|
| Involvement in PBT * | x ** | x | x | 5 | 5 | 3 | 3 |
| Satisfied with PBT | 5 | 5 | 5 | 5 | 5 | 5 | 4 |
| Extent PBT successful for child's participation | 3 | 5 | 5 | 5 | 5 | 5 | 4 |
| Extent communicative | | | | | | | |
|     Physical therapist | 5 | 4 | 5 | 5 | 5 | 5 | 4 |
|     Family | x | x | 5 | 2 | 5 | 5 | 3 |
|     Teacher | 5 | 4 | x | 5 | 5 | 5 | 4 |
|     Other IEP *** members | 2 | 5 | 5 | x | x | x | x |

**Table 6.** *Cont.*

| Question | Jake's Mother | Luke's Mother | Teacher | PT | OT | SLP | Vision |
|---|---|---|---|---|---|---|---|
| Extent IEP team collaborative | 4 | 5 | 5 | 5 | 5 | 5 | 4 |
| Extent PBT enhanced | | | | | | | |
|     Collaboration | 4 | 5 | 5 | 5 | 5 | 5 | 4 |
|     Confidence | 4 | 5 | 5 | 5 | 5 | 3 | 4 |
| Extent PBT promoted child's self determination | 3 | 4 | 5 | 5 | 5 | 4 | 4 |
| Extent child enjoyed participation | 5 | 4 | 5 | 5 | 5 | 5 | 4 |
| Helpful | | | | | | | |
|     ID **** goal for participation and GAS | 4 | 4 | 5 | 5 | 5 | 3 | 4 |
|     ID strengths/outcomes | 4 | 4 | 5 | 5 | 5 | 4 | 3 |
|     Meeting for action plan | 4 | 5 | 5 | 5 | 5 | 4 | 4 |
|     Developing action plan | 4 | 4 | 5 | 5 | 5 | 4 | 4 |
|     Implementing action plan | 3 | 4 | 5 | 5 | 5 | 5 | 4 |
|     Sharing videos with family | 5 | 5 | 5 | 5 | 5 | 0 | 4 |
|     Sharing videos with PT ***** and Teacher | 4 | 5 | 5 | 5 | 5 | 0 | 4 |
| PT and teacher collaborating to support child participation | 4 | 5 | 5 | 5 | 5 | 5 | 4 |
| PT collaborating with related service providers | 4 | 5 | 5 | x | 5 | 5 | 4 |

Note: * PBT = participation-based therapy; ** x = question not asked of participant; *** IEP = Individualized Education Program; **** ID = identified; ***** PT = physical therapist; Scale: 5 = *completely satisfied/very great extent/extremely helpful/completely/involved*; 4 = *very satisfied/great extent/very helpful/very involved*; 3 = *moderately satisfied/moderate extent/somewhat helpful/moderately involved*; 2 = *slightly satisfied/small extent/slightly helpful/slightly involved*; 1 = *not at all satisfied/not at all/not at all helpful/not at all involved*; 0 = *unsure*.

## 4. Discussion

This case series enabled a rich description of the complexity of adapting the participation-based approach in the preschool educational setting for young students with multiple disabilities and the reporting of the experiences of all members of the IEP team. The findings suggest that a participation-based therapy approach enabled the preschool IEP team to have meaningful collaborations, allowing for high engagement in the process, which strengthened the family–school partnership. This approach utilized team collaborative goal development and service planning and implemented holistic participation-based interventions, which contributed to positive student outcomes and team experiences.

The collaborative meetings for goal development highlighted the mothers' priorities for physical recreation, and the teacher and PT were able to contextualize the goals for the educational environment and ensure alignment with the Pennsylvania Learning Standards for Early Childhood [28]. This collaborative process was novel to the IEP teams as, previously, the IEP team would discuss each student's present level of performance and develop a goal in alignment with a desire for skill progression [29]. The goal development meetings enabled the team to discuss activities that reflected student participation integrated across educational domains versus discrete goals by individual disciplines [30]. The finding that the mothers, teacher, and PT rated the goal development process as "very to extremely helpful" supports the value of this process and is consistent with Hunt and colleagues' [6] report that collaborative processes enabled preschool team members to have a more holistic view of the student. The variability in the ratings for the helpfulness of identifying the goal by the RSPs ("somewhat helpful" to "extremely helpful") could be related to the goal being centered around physical recreational activity or the need to include all RSPs in goal development, as only the PT was in the goal selection meetings. Although efforts were made to identify goal activities that the students might enjoy, the use of innovative approaches to solicit students' preferences should be considered in the future [31]. We recommend including all IEP team members in goal selection and the identification of

GAS levels to enhance team collaboration, the family–school partnership, and student self-determination.

Trusting family–school partnerships were established during the *Collaborative Process for Action Plans* meetings, which were valued by all team members. These meetings enabled the members to listen to each other's perspectives on the attributes influencing the goal and develop an action plan collectively. During the meetings, the mothers portrayed a desire to share specific information on their child's strengths and needs. Turnbull and colleagues [32] identified trust as the founding principle of the family–professional partnership. The implementation of the participation-based approach highlighted that the families' trust in the school-based members varied. Jake's family received virtual school services during the COVID-19-mandated closure of schools and was at home with a nurse; therefore, his mother acclimated to the support of others and appeared more trusting of the school-based team. Contrary to this, Luke's family did not participate in virtual sessions, as the mother was concerned that the school-based team would not understand her son's strengths and needs. The meeting enabled Luke's mother to gain trust in the school-based team's competencies and investment in her son. The PT perceived the meetings as the best part of the participation-based therapy approach because the discussion was from a strengths-based perspective and the mothers were equal partners in the conversation.

The action plans were detailed and comprehensive, providing actionable items for specific members of the IEP teams to meet each student's goal; however, the written document was not shared with each member after the meeting. Although, via the experience questionnaires, the IEP teams reported that the development of the action plan was "very helpful" to "extremely helpful", the ratings for the implementation of the action plan varied from "somewhat helpful" to "extremely helpful". This may be attributable to the lack of sharing the final "collaborative action plan" and may reflect a need for more detailed communication to all members of the IEP team regarding the interventions that were being provided at school, and this was particularly reflected in Jake's mother's responses. Unlike community-based interventions that support parents to foster their children's physical activity [33], efforts need to be made by school-based teams to facilitate parental involvement. Communication among all team members was particularly important during COVID-19 as the school-wide health plan limited the number of RSPs that could be present in the classroom, caused students and staff to frequently quarantine, and necessitated the use of masks, which hindered nonverbal communication. Hunt et al. [6] suggested that effective collaboration includes building accountability into the processes when implementing supports and holding regular meetings to review their effectiveness. While the "*Collaborative Process for Action Plans*" encourages collaboration, sharing the written document would have built in accountability for the supports and interventions and facilitated more effective collaboration.

Communication using technology requires time and an intentional approach to promote engagement. Videos were often sent without a narrative explanation, and this may not have been an effective method to enable collaboration. Though the classroom teacher and families described the sharing of videos as "very helpful" to "extremely helpful", the PT rated the extent to which the families were communicative as "small extent", as families did not often respond to texts or videos or share information related to their child's participation at home. Video sharing could be included as a part of team meetings in which members share and discuss videos that demonstrate the interventions, the students' skills, and their progress toward goal achievement. The National Association for the Education of Young Children's statement on developmentally appropriate practices [4] highlights the importance of frequent, reciprocal communication and the need to use a variety of communicative means (informal and formal conversations, phone calls, texting, and emails), based on family preference, to effectively collaborate with families.

Providing interventions during the classroom activities enabled the team to share their expertise, fostered engagement, and supported the classroom staff's confidence to utilize the strategies independently. These school-based team interactions are aligned with

community-based initiatives that have recognized the role of strength-based attitudes, supports, and relationships as essential ingredients [34]. The team demonstrated solution-focused strategies to facilitate student participation with their peers despite the pandemic restrictions. The accessibility of the recreation equipment, the joyful engagement of the students, and modeling holistic participation-based interventions motivated the school-based team to engage the children in the activities when the PT was not present. The students' joyous responses to the activities were uplifting to the whole class, fostering a positive energy in the classroom [35]. This was especially true for Luke, who would stop crying to play the bowling activity. The high ratings of satisfaction with the participation-based therapy program aligned with the students' success in goal attainment.

### 4.1. Limitations and Recommendations for Future Research

The outcomes of this study were directly related to the students, the dynamics of the team, and the environment. Due to it being a case series, the findings of this research study are not generalizable. The established relationship between the PT and the other school-based team members made it possible to implement this study but also may have led to some participant bias. The environment offered adapted recreation equipment that may not be readily available in all preschools. An inherent bias in conducting case-based research exists with the researcher also being the participants providing services. To decrease research bias, a rigorous prospective protocol that included the bracketing of assumptions prior to the start of the study, the journal-based documentation of reflections, and gathering data from multiple sources was utilized. In future, a cluster randomized study is needed to examine the effects of participation-based therapy on team collaboration, student engagement, and goal attainment.

### 4.2. Implications for Practice

A participation-based therapy approach may support the development of meaningful transdisciplinary goals, collaboration within IEP teams, holistic embedded interventions, trusting collaborative family–school partnerships, and student goal attainment. Goals written holistically, describing the task across physical, behavioral, communicative, and cognitive components, may make them more salient to the team. GAS can be utilized in educational environments and is beneficial for progress monitoring. Action plans developed collaboratively may offer more detailed delineations of what strategies need to be utilized to foster accountability and promote student goal achievement. The use of technology to hold virtual meetings may decrease barriers to attendance and enable more frequent communication and collaboration regarding interventions and progress toward goals. Personalizing the technology for families and adhering to their technological and communication preferences may encourage more reciprocal communication.

### 5. Conclusions

Based on the outcomes of this study, participation-based therapy was deemed feasible for use in a preschool educational setting for two students with multiple disabilities and was acceptable according to the IEP teams. Key considerations for successful implementation include dedicating time to allow for collaboration among all team members and the development of trusting relationships and solution-focused participation-based interventions.

**Supplementary Materials:** The following supporting information can be downloaded at: https://www.mdpi.com/article/10.3390/disabilities3040038/s1, Supplement S1: The *Collaborative Process for Action Plans to Achieve Participation Goals*; Supplement S2: Intervention log; Supplement S3: Student engagement questionnaire; Supplement S4: Participant experience questionnaire sample; Supplement S5: Student IEP goals; Supplements S6 and S7: The *Collaborative Process for Action Plans to Achieve Participation Goals:* analysis of attributes; Supplement S8: Participants' responses to open-ended items on experience questionnaire.

**Author Contributions:** Conceptualization, H.L.B. and L.A.C. methodology, H.L.B., L.A.C., R.J.P. and K.D.W.; formal analysis, H.L.B. and L.A.C.; investigation, H.L.B. and L.A.C.; writing—original draft preparation, H.L.B., L.A.C., R.J.P. and K.D.W.; writing—review and editing, H.L.B., L.A.C., R.J.P. and K.D.W. All authors have read and agreed to the published version of the manuscript.

**Funding:** This work received no external funding.

**Institutional Review Board Statement:** This study was conducted in accordance with the Declaration of Helsinki and approved by the Institutional Review Board of Drexel University (protocol code 2009008077, 10 May 2020).

**Informed Consent Statement:** Informed consent was obtained from all subjects involved in the study. Written informed consent has been obtained from the patient(s) to publish this paper.

**Data Availability Statement:** The data presented in this study are not available due to participant privacy.

**Acknowledgments:** This research was completed in partial fulfillment of Heather Brossman's Doctor of Health Science in Rehabilitation Sciences (DHSc) at Drexel University.

**Conflicts of Interest:** The authors declare no conflict of interest.

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
