# Peer review of "A Collaborative Approach to Support Participation in Physical Recreation for Preschool Students with Multiple Disabilities: A Case Series"

_disabilities, doi:10.3390/disabilities3040038_

Round 1

Reviewer 1 Report

Comments and Suggestions for Authors

Overall:

Thank you for the opportunity to review this interesting paper outlining an evaluation of a collaboration process for action plans to achieve participation goals intervention for 2 preschool age children, presented as a case series.

This paper outlines a comprehensive approach for improving PA participation in an educational setting. It provides a unique perspective, especially around individual interventions in preschool settings.

While this paper provides wonderful detail on what the intervention looked like and what was involved in developing goals and strategies (the supplementary materials were excellent for this), I would like to see more information on HOW the intervention was evaluated. This will give readers more confidence in your conclusions.

Specifically:

The methods should include some sort of details as to how the outcome measures were evaluated and interpreted. This could just be as simple as indicating you are reporting results as descriptive statistics. As you have open ended questions, you do need to give some indication as to what was done with this data. I do realise it will be a small amount of data for a case series of n=2, but including quotes without any indication as to what was done with the rest of the data doesn’t paint a complete picture.

I also found the formatting of your tables challenging to read, I’ve put some more specific comments below.

Introduction:

Clearly outlines current evidence base and research question.

A few small comments:

Line 71. Consider some more recent participation approaches- there are lots of interesting ones coming from Canada, Australia and the Nordic countries.  

Line 75-77. This reads a little awkwardly. Do you mean that previous research has not considered a) roles of family, educators and therapists, and b) hasn’t explored PA in a preschool education program setting?

It’s worth noting that there are a few participation-focused interventions that have evaluated or reported experiences of healthcare professionals, family members etc. eg. Willis et. Al. 2020 https://doi.org/10.1080/09638288.2021.1907458, or Clutterbuck et. Al. 2020 https://search.ebscohost.com/login.aspx?direct=true&AuthType=sso&db=mnh&AN=32633156&site=eds-live&scope=site.

Methods:

There is quite a large amount of identifying information included in this case study. I just wanted to double check that the families are aware of what information is included in the paper?

Table 3 and 4.

I find the formatting of this table difficult to understand. It is not easy to read who is responsible to what, and the centre format of the dot points makes it generally difficult to read.

Table 5

 It needs to be clearer what the percentages are referring to. Sometimes this is indicated in the description, other times it is not.

3.2.1

In the methods, it states that the student engagement questionnaire has 2 Likert scale questions and four open ended questions.

Are there any results for the Likert scale questions?

How were the open-ended questions analysed? I realise a small number of open-ended questions won’t generate a lot of data, but sharing how you managed the responses to these questions will improve the confidence of your readers that you haven’t just chosen the most positive things reported by the educators.

3.2.2

Are there any findings from the open-ended questions from the satisfaction questionnaire?

If there are, please include some information about how the open-ended data was analysed.

Table 6.

I found this table quite difficult to interpret. Please include some information about what the number mean (I’m guessing ‘Likert scale response (1= strongly disagree, 5=strongly agree)’ or something like that?)

Could you also consider including the complete question? For example is ‘confidence’ referring to the child’s confidence or the respondents confidence in their role in the participation intervention? This would make the findings much clearer for readers.

Discussion

The summary of findings in the discussion are a little hard to put in context, as the findings from the students engagement questionnaire and the participant experience questionnaire weren’t clear to me in the results section. Furthermore, the discussion appears to include results (such as perceptions of parents) that weren’t included in the results section. Furthermore, in your discussion, you’ve used quotes from parents etc. that illustrate nicely the strengths of the program. Where are these quotes from? Are they from one of the outcome measures?

I think some additional information and reporting in your results section will make the discussion section stronger, as at the moment it just feels like I am missing some of the results you are discussing. 

Supplementary materials.

Thank you for including these- very helpful and detailed.

Just a suggestion, but including the Student Engagement questionnaire and the Experience Questionnaires as supplementary would be very helpful in understanding your results in more detail.

Comments on the Quality of English Language

Two things:

Use of past tense describing your two participants is quite off-putting to me, though not grammatically incorrect (I don't think). Could I suggest:

Jake: At the time of the intervention, Jake was 3.5 years old. Jake is (then continue in present tense).  

Not English language- but the table formatting for all tables are quite difficult to read. Tables are usually left aligned. And it would be helpful is subheadings etc. were differentiated in some way (bold, italics, line divides etc.) 

Author Response

Response to Reviewer 1 Comments

Comment: I would like to see more information on HOW the intervention was evaluated. This will give readers more confidence in your conclusions.

Response: I appreciate this comment. Our interventions were evaluated in two ways, detailed in the procedure section 2.1.2.  One assessment of intervention was through the determination of goal attainment scaling with both the physical therapist and teacher. The second evaluation was the experience questionnaires that each participant completed to document the evaluation of this process. Section 2.1.3 has been added to end of methods’ section to explain the data synthesis.  

Comment: Introduction: Line 71. Consider some more recent participation approaches- there are lots of interesting ones coming from Canada, Australia and the Nordic countries.  Line 75-77. This reads a little awkwardly. Do you mean that previous research has not considered a) roles of family, educators and therapists, and b) hasn’t explored PA in a preschool education program setting? It’s worth noting that there are a few participation-focused interventions that have evaluated or reported experiences of healthcare professionals, family members etc. eg. Willis et. Al. 2020 https://doi.org/10.1080/09638288.2021.1907458, or Clutterbuck et. Al. 2020 https://search.ebscohost.com/login.aspx?direct=true&AuthType=sso&db=mnh&AN=32633156&site=eds-live&scope=site.

Response: We agree there are other participation approaches. Law et al., reference 12 in our article, is from the Canada group. However, we realize this does not reflect this groups’ current participation framework, which was cited in the next sentence (Law et al., 2015). We have now included that reference in line 71 along with Willis et al. 2018 (Australia and Nordic countries) and Reedman et al. 2019 (Australia). The work from the Finland group (Vanska et al, 2020) on meaningful participation from the perspective of children has now been referenced in our discussion section. We have now clarified the gap in the research. (Lines 75-79).

Comment: Methods: There is quite a large amount of identifying information included in this case study. I just wanted to double check that the families are aware of what information is included in the paper?

Response: The cases used pseudonyms.  The study was approved by Drexel IRB as indicated. The two families gave informed consent which included reference to potential publication of the study.

Comment: Methods: The methods should include some sort of details as to how the outcome measures were evaluated and interpreted. This could just be as simple as indicating you are reporting results as descriptive statistics. As you have open ended questions, you do need to give some indication as to what was done with this data. I do realize it will be a small amount of data for a case series of n=2 but including quotes without any indication as to what was done with the rest of the data doesn’t paint a complete picture.

Response: Thank you for this feedback. As indicated above, we have now added a section on Data Synthesis in the Methods. (Starting on Line 153)

Comment: Formatting tables challenging to read. Table 3 and 4: I find the formatting of this table difficult to understand. It is not easy to read who is responsible to what, and the center format of the dot points makes it generally difficult to read. Table 5: It needs to be clearer what the percentages are referring to. Sometimes this is indicated in the description, other times it is not. the table formatting- all tables are quite difficult to read. Tables are usually left aligned. And it would be helpful is subheadings etc. were differentiated in some way (bold, italics, line divides etc.) 

Response: Agreed. The formatting apparently changed when put into submission.  It was not intended to be centered. Thank you for the additional points of clarification. The tables were reorganized and edited.

Comment: 3.2.1 In the methods, it states that the student engagement questionnaire has 2 Likert scale questions and four open ended questions. Are there any results for the Likert scale questions?

Response: The descriptive results from the Likert-scaled questions were provided in the narrative outcomes. We have now clarified that outcome, added the response scale to the measure description, and included the questionnaire as a supplement.

Comment: How were the open-ended questions analyzed? I realize a small number of open-ended questions won’t generate a lot of data, but sharing how you managed the responses to these questions will improve the confidence of your readers that you haven’t just chosen the most positive things reported by the educators.

Response: We have now included a statement in the data synthesis section (2.1.3) to describe how the open-ended responses were analyzed, noting that both positive and constructive feedback was represented. In this regard we have now expanded the narrative on the findings from the student engagement questionnaires, such as adding the teacher’s report of the students’ dislikes about participating in the activity, have included a supplemental table (supplement 8) of the open-ended responses from the experience questionnaires, and have expanded the narrative highlighting these findings.

Comment: 3.2.2 Are there any findings from the open-ended questions from the satisfaction questionnaire? If there are, please include some information about how the open-ended data was analyzed.

Response: As noted above, a statement on how the open-ended responses were analyzed has been added. The findings from the open-ended questions on the participants’ experience questionnaires have been included in Supplement 8 and highlighted in the narrative.

Comment: Table 6 I found this table quite difficult to interpret. Please include some information about what the number mean (I’m guessing ‘Likert scale response (1= strongly disagree, 5=strongly agree)’ or something like that?)

Response: In transmission the table footnote was lost and has now been re-inserted which includes the abbreviations and scale response options.

Comment: Could you also consider including the complete question? For example, is ‘confidence’ referring to the child’s confidence or the respondent’s confidence in their role in the participation intervention? This would make the findings much clearer for readers.

Response: The questionnaire has been included with supplemental materials.

Comment: Discussion -The summary of findings in the discussion are a little hard to put in context, as the findings from the student’s engagement questionnaire and the participant experience questionnaire weren’t clear to me in the results section. Furthermore, the discussion appears to include results (such as perceptions of parents) that weren’t included in the results section. Furthermore, in your discussion, you’ve used quotes from parents etc. that illustrate nicely the strengths of the program. Where are these quotes from? Are they from one of the outcome measures? I think some additional information and reporting in your results section will make the discussion section stronger, as at the moment it just feels like I am missing some of the results you are discussing. 

Response: The findings from the student’s engagement questionnaire have been clarified in the narrative of the outcomes’ section. A supplemental table (supplement 8) has been added to the outcomes to present the findings from the open-ended questions on the participant experience questionnaires including sample quotes. Quotes have been reported in the outcome section and have been removed from the discussion.

Comment: Supplementary materials. Thank you for including these- very helpful and detailed. Just a suggestion but including the Student Engagement questionnaire and the Experience Questionnaires as supplementary would be very helpful in understanding your results in more detail.

Response: See responses above, we have now included these materials

Comment-Two things. Use of past tense describing your two participants is quite off-putting to me, though not grammatically incorrect (I don't think). Could I suggest: Jake: At the time of the intervention, Jake was 3.5 years old. Jake is (then continue in present tense).  

Response: Thank you for the feedback. We made these edits to the participant’s description.

Reviewer 2 Report

Comments and Suggestions for Authors

Dear Authors,

your contribution is highly interesting and very well described. There are only minor comments from my side:

- no ethics check is given. I am aware that reviews by ethics committees are very challenging in this research area. Nevertheless, it is indispensable to include a reference to ethical aspects here or to refer to the ethics review.
- the discussion sounds partwise like adding information to the your results. It lacks a part on how the results fit into the current state of research. The Introduction mentions that there is little research on this and shows sources from 2004 and 2017. It is good to describe if anything has happened between the years, if there really is not more research on this since 2017. Maybe what are others results and what is the benefit of your results. This could better highlight the importance of one's findings.

Author Response

Response to Reviewer 2 Comments

Comment: no ethics check is given. I am aware that reviews by ethics committees are very challenging in this research area. Nevertheless, it is indispensable to include a reference to ethical aspects here or to refer to the ethics review.

Response: This information was provided in the first sentence of section 2.2.1 participants and setting. That paragraph also denotes that informed consent was obtained.

Comment: The Introduction mentions that there is little research on this and shows sources from 2004 and 2017. It is good to describe if anything has happened between the years, if there really is not more research on this since 2017.

Response: Additional references have been added to the introduction.

Comment: the discussion sounds partwise like adding information to your results. It lacks a part on how the results fit into the current state of research. Maybe what are others results and what is the benefit of your results. This could better highlight the importance of one's findings.

Response: The quotes have been removed from the discussion and added to the outcomes’ section. We have clarified the gap in the literature in the introduction and have more explicitly highlighted the contribution of this case series in the first paragraph of the discussion.